# Aetiology of lobar pneumonia determined by multiplex molecular analyses of lung and pleural aspirate specimens in the Gambia: findings from population-based pneumonia surveillance

Grant Austin Mackenzie [ID],[1,2,3] Jessica McLellan,[1,4] Eunice Machuka,[1] Malick Ndiaye,[1] Jayani Pathirana,[1] Augustin Fombah,[1] Baderinwa Abatan,[1] Ilias Hossain,[1] Ahmed Manjang,[1] Brian Greenwood,[3] Philip Hill[5]

**Correspondence to**
Dr Grant Austin Mackenzie; gmackenzie@mrc.gm

## ABSTRACT

**Objectives** To determine the causes of lobar pneumonia in rural Gambia.

**Design and setting** Population-based pneumonia surveillance at seven peripheral health facilities and two regional hospitals in rural Gambia. 7-valent pneumococcal conjugate vaccine (PCV7) was introduced routinely in August 2009 and replaced by PCV13 from May 2011.

**Methods** Prospective pneumonia surveillance was undertaken among all ages with referral of suspected pneumonia cases to the regional hospitals. Blood culture and chest radiographs were performed routinely while lung or pleural aspirates were collected from selected, clinically stable patients with pleural effusion on radiograph and/or large, dense, peripheral consolidation. We used conventional microbiology, and from 8 April 2011 to 17 July 2012, used a multiplex PCR assay on lung and pleural aspirates. We calculated proportions with pathogens, associations between coinfecting pathogens and PCV effectiveness.

**Participants** 2550 patients were admitted with clinical pneumonia; 741 with lobar pneumonia or pleural effusion. We performed 181 lung or pleural aspirates and multiplex PCR on 156 lung and 4 pleural aspirates.

**Results** Pathogens were detected in 116/160 specimens, the most common being *Streptococcus pneumoniae* (n=68)*, Staphylococcus aureus* (n=26) and *Haemophilus influenzae* type b (n=11). Bacteria (n=97) were more common than viruses (n=49). Common viruses were bocavirus (n=11) and influenza (n=11). Coinfections were frequent (n=55). *Moraxella catarrhalis* was detected in eight patients and in every case there was coinfection with *S. pneumoniae*. The odds ratio of vaccine-type pneumococcal pneumonia in patients with two or three compared with zero doses of PCV was 0.17 (95% CI 0.06 to 0.51).

**Conclusions** Lobar pneumonia in rural Gambia was caused primarily by bacteria, particularly *S. pneumoniae* and *S. aureus*. Coinfection was common and *M. catarrhalis* always coinfected with *S. pneumoniae*. PCV was highly efficacious against vaccine-type pneumococcal pneumonia.

## Strengths and limitations of this study

► Population-based pneumonia surveillance collecting gold standard specimens directly from the infected lung to determine the aetiology of lobar pneumonia.

► Multiplex real-time quantitative PCR was used to detect up to 31 pathogens in lung specimens.

► However, multiplex PCR excluded *Legionella*, *Klebsiella* and *Mycobacterium tuberculosis* and there was failure to detect a pathogen in 28% of patients with a lung specimen.

► Specific aetiology results and accurate vaccination records allowed calculation of the effectiveness of pneumococcal conjugate vaccine to prevent non-bacteraemic pneumococcal pneumonia.

► Results are generalisable to patients with lobar pneumonia, but not to all patients with clinical pneumonia.

## INTRODUCTION

The aetiology of childhood pneumonia is difficult to determine for a number of reasons: the upper respiratory tract is often colonised by pneumonia pathogens, a problem exacerbated with the use of overly sensitive molecular methods, the inability to produce sputum of good quality, and the difficulty obtaining a specimen from the alveolar space. Most studies of the aetiology of pneumonia rely on either the insensitive culture of bacteria from blood or the non-specific detection of organisms in sputum or pharynx. Case-control studies have compared the prevalence of organisms in the pharynx of

children with pneumonia and matched controls, relying on the assumption that organisms detected in the pharynx are also present and pathogenic in the lung.[1–4] The multi-site Pneumonia Etiology for Research in Child Health (PERCH) study extended these methods, combining conventional and molecular microbiology data from the pharynx, blood, and lung with an analytic approach to estimate the probability of specific aetiologies.[2]

Historic studies using lung aspirate specimens and conventional microbiology commonly found *Streptococcus pneumoniae* and *Haemophilus influenzae* to be the most frequent causes of lobar pneumonia.[5–8] More recent studies using lung aspirates have been uncommon. A Gambian study employing molecular methods in 47 lung and nine pleural aspirates, and the PERCH study with 37 lung and 15 pleural aspirates, identified a pneumococcal aetiology in 87% and 25% of patients, respectively.[2 9] Coinfection was present in 51%[9] and 17% of patients, respectively.[2 9] The PERCH study may have underestimated the prevalence of bacterial infection in pneumonia due to the inclusion of children with bronchiolitis, challenges enrolling very sick children, and an assumption that organisms in pharyngeal specimens correlate with the cause of pneumonia.[10]

The importance of determining the aetiology of pneumonia, particularly the role of coinfections and the impact of vaccination strategies, remains. We studied these questions during the introduction of pneumococcal conjugate vaccination (PCV), applying conventional and molecular methods to lung specimens. We aimed to determine the aetiology of lobar pneumonia and the effectiveness of PCV to prevent pneumococcal pneumonia in rural Gambia.

## METHODS
### Setting
This study was nested within a population-based surveillance study for suspected pneumonia, septicaemia or meningitis in the Basse and Fuladu West Health and Demographic Surveillance Systems (BHDSS and FWHDSS) in rural Gambia, which in January 2012, included approximately 170 043 and 89 389 residents, respectively. Child mortality in the BHDSS in 2011 was 68 per 1000 live births. Surveillance commenced in the BHDSS on 12 May 2008 and in the FWHDSS on 12 September 2011. PCV7 was introduced on 19 August 2009 and replaced by PCV13 during May 2011.

### Patients and procedures
The surveillance has been described previously.[11] All patients presenting to the nine health facilities in the study area were screened 7 days per week, 24 hours per day, using standardised methods to detect possible cases of pneumonia, septicaemia, meningitis, referral and clinical investigation.[12 13] We used cross-sectional enrolment including all patients aged ≥2 months with suspected pneumonia. Suspected pneumonia was defined using

modified WHO criteria, as a history of cough or difficulty breathing with the presence of any one of the following: respiratory rate ≥40 or ≥50 per minute for children aged greater than or less than 12 months respectively, lower-chest-wall-indrawing, nasal flaring, grunting, oxygen saturation <92%, dullness to percussion, bronchial breathing or crackles on auscultation. Patients with suspected pneumonia had anthropometric measurements, peripheral oxygen saturation measured, blood cultured and chest radiographs done. We did not test for HIV as this was not standard practice and prevalence in The Gambia is relatively low.[14] Chest radiographs were interpreted according to WHO recommendations[15] by two independent reviewers, with readings discordant for end-point consolidation (ie, lobar pneumonia) resolved by a third reviewer. A percutaneous trans-thoracic lung or pleural fluid aspiration was considered if a pleural effusion or large, dense, peripheral pneumonic consolidation was present on radiograph, there were no contraindications (postmeasles pneumonia, pneumatocoeles on radiograph, skin sepsis or no written informed consent), and the patient was clinically stable. Following written, informed consent, lung aspiration was performed by a clinician using aseptic technique with a 21 gauge needle and 5 mL syringe with 1 mL of sterile saline with an aliquot inoculated on culture media. Specimens were immediately transported to the MRC Gambia, Basse laboratory, for preliminary analysis and stored at −80°C. Patients were observed for 3 hours post-procedure. Lung aspiration is established as a safe practice in The Gambia, with an excellent safety record and sensitivity as a diagnostic tool.[16] All patients admitted with clinical pneumonia from 8 April 2011 to 17 July 2012 were included in this study. We chose this period as it covered the introduction period of PCV.

### Laboratory procedures
Microbiological specimens were processed in Basse using conventional microbiological methods including staining of lung and pleural aspirates for *Mycobacterium tuberculosis*.[17] Blood was cultured using an automated system (Bactec 9050, Beckton Dickinson, Belgium). Microbiological results were used to inform patient care. We serotyped *S. pneumoniae* isolates by latex agglutination using factor and group-specific antisera (Statens Serum Institute, Copenhagen, Denmark).[18] *H. influenzae* isolates were serotyped by slide agglutination using polyvalent and monovalent antisera to types a, b, c, d, e and f (Beckton Dickinson, Erembodegem, Belgium). Isolates that did not agglutinate with polyvalent antisera were classified as non-typeable *H. influenzae*.

Molecular analysis of lung specimens was conducted in two batches, in November/December 2011 and 2012, using the same methods, staff and laboratory as in the PERCH study in The Gambia. Total nucleic acid was extracted from a 200 µL aliquot of lung and pleural aspirates (easyMAG, bioMérieux, France) with an internal control. Extracts were subjected to quantitative multiplex PCR (Fast-track Diagnostics Resp-33 kit, Sliema, Malta) for

a panel of 33 respiratory bacteria, fungi, and viruses (see online supplemental material) with internal positive, and negative controls.[19] Standard PCR curves were derived from plasmid standards during the testing to calculate pathogen load from cycle threshold values. We did not use a density threshold to define a positive result based on the assumption that any putative pathogen detected in consolidated lung or pleural fluid is pathogenic and involved in the pneumonic process. Interpretation for some targets required combinations of results(see online supplemental material). Assay specificity for the *Klebsiella pneumoniae* and *Legionella* spp targets was poor and therefore results for these bacteria were omitted from analyses.

## Statistical analysis

We summarised the characteristics of patients admitted with clinical pneumonia and classified them into three groups; no radiological lobar consolidation and lobar consolidation with or without lung or pleural aspirate. Categorical variables were assessed using $\chi^2$ tests and the Kruskal-Wallis test was used for continuous variables. We calculated age-stratified proportions of patients with pathogens identified in lung or pleural aspirates using multiplex PCR. Values of pathogen quantity were transformed to $\log_{10}$ copies per ml. We tabulated the frequency of coinfection by pairs of pathogens. We used test-negative analyses to estimate the effectiveness of PCV to prevent pneumococcal pneumonia and vaccine-type pneumococcal pneumonia; combining conventional culture and serotype results with PCR results as appropriate. We calculated the odds of a positive vs negative test for the outcome in patients who had received ≥2 doses of PCV compared with zero doses seven or more days before admission. We calculated odds ratios and 95% CIs in crude and age-stratified analyses using the Mantel-Haenszel method. Fisher's exact p values were used for hypothesis tests. Analyses were done using STATA V.16 (StataCorp).

## Patient and public involvement

Patients and public were not involved in the design and conduct of the primary surveillance study that generated the data analysed for this report. Reporting of the primary study results and dissemination of results included a joint press release by the Gambia Ministry of Health and the Medical Research Council Unit The Gambia at London School of Hygiene & Tropical Medicine (MRCG at LSHTM), as well as local feedback to health authorities and local communities in the study area. Patients and public were not involved in the specific sub-analysis of pneumonia aetiology data presented in this manuscript.

## RESULTS

Over the 21-month study period from 8 April 2011 to 17 July 2012, 2550 patients were hospitalised with clinical pneumonia; 2406 were aged 0–59 months and 141 were aged ≥5 years (figure 1). WHO-defined radiological

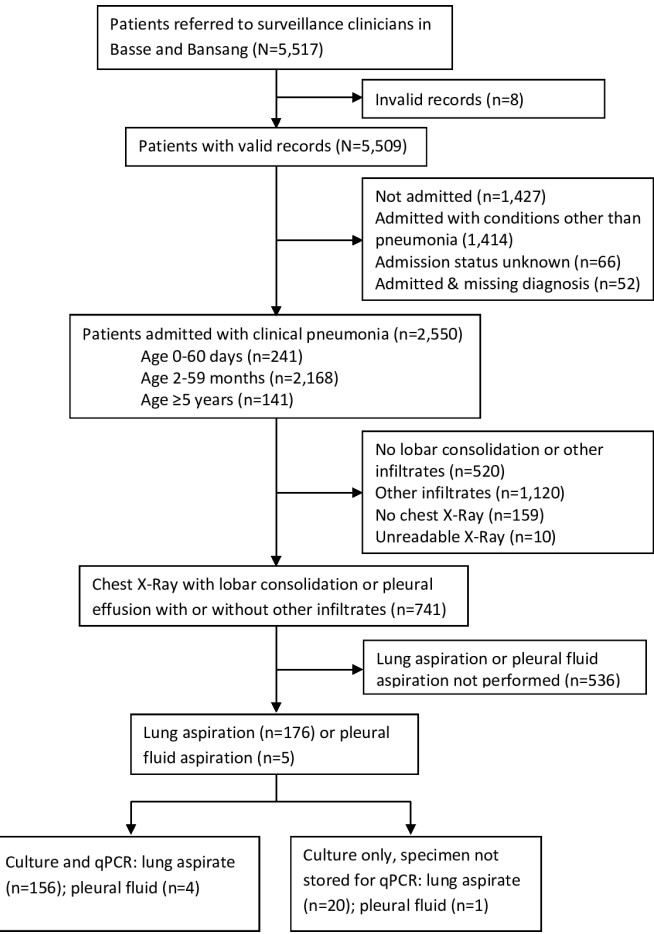

**Figure 1** Study profile.

pneumonia with consolidation(ie, lobar pneumonia) was detected in 741 (29%) patients. Of those with lobar pneumonia, lung or pleural aspirates were collected from 176 and five (24%, 181/741) patients respectively. There were no complications following the lung aspiration procedures. Patients with lobar pneumonia aged 0–60 days were less likely than older patients to have a lung aspirate (1/64 vs 180/681) while older children and adults were more likely to have a lung aspirate than children aged 2–59 months (44/89 vs 136/592) (table 1). Bacteraemia was more common in patients who had a lung aspirate (31/178, 17%) compared with those without a lung aspirate (113/2119, 5%).

Multiplex PCR was performed on 160/181 lung and pleural aspirates. Twenty-one collected specimens were not stored or available for PCR analysis. Before the exclusion of *K. pneumoniae* and *Legionella* results due to poor specificity, at least one pathogen was detected in 132/160 patients, and after their exclusion, pathogens were detected in 116/160 (73%) lung specimens (lung and pleural aspirates combined), one pathogen in 61 (38%) and two or more in 55 (34%) (table 2). Bacteria were detected in 97 (61%) specimens and viruses in 49 (31%). Bacteria only infections were detected in 67 (42%) and bacterial coinfections in 26 (16%) specimens. Viral only

**Table 1** Characteristics of 2550 patients admitted to hospital with clinical pneumonia, radiological findings and investigation with lung or pleural aspiration

| Characteristic | Subgroup | No lobar consolidation (N=1646) | Lobar consolidation no lung/pleural aspirate (N=564) | Lobar consolidation and lung/pleural aspirate (N=181) |
|---|---|---|---|---|
| Age | 0–60 days | 143 (8.7%) | 63 (11.2%) | 1 (0.5%) |
| | 2–59 months | 1452 (88.2%) | 456 (80.9%) | 136 (75.1%) |
| | 5–14 years | 42 (2.6%) | 23 (4.1%) | 26 (14.4%) |
| | ≥15 years | 9 (0.5%) | 22 (3.9%) | 18 (9.9%) |
| Male | | 933 (56.7%) | 324 (57.4%) | 104 (57.5%) |
| Mean respiratory rate/min* | | 57.3 | 61.6 | 60.8 |
| Mean oxygen saturation %* | | 95.8% | 93.6% | 95.1% |
| Wheeze* | | 319/1641 (19.4%) | 76/562 (13.5%) | 11/181 (6.1%) |
| Tachycardia† | | 963 (58.5%) | 351 (62.2%) | 138 (76.2%) |
| Temperature ≥38°C* | | 823 (50.0%) | 347 (61.5%) | 126 (69.6%) |
| Prostration*‡ | 0–59 months | 116/1572 (7.4%) | 34/513 (6.6%) | 5/136 (3.7%) |
| WfH z-score <-3* | 0–59 months | 274/1587(17.3%) | 95/513 (18.5%) | 28/136 (20.6%) |
| BMI grade three thinness | 5–17 years | 9/44 (20%) | 8/28 (29%) | 8/28 (29%) |
| BMI <18.5 kg/m² | ≥18 years | 1/7 (14%) | 4/16 (25%) | 3/15 (20%) |
| Blood culture taken | | 1584 (96.2%) | 535 (94.9%) | 178 (98.3%) |
| Blood culture pathogen isolated | | 82/1584 (5.2%) | 31/535 (5.8%) | 31/178 (17.4%) |
| PCV immunisation doses§ | 0 | 357/1452 (24.6%) | 109/456 (23.9%) | 43/136 (31.6%) |
| | 1 | 159/1452 (11.0%) | 38/456 (8.3%) | 12/136 (8.8%) |
| | 2 | 152/1452 (10.5%) | 50/456 (11.0%) | 10/136 (7.4%) |
| | 3 | 784/1452 (54.0%) | 259/456 (56.8%) | 71/136 (52.2%) |
| Died in hospital | | 65 (3.9%) | 25 (4.4%) | 6 (3.3%) |

Column totals do not equal 2550 as 159 patients did not have a chest radiograph.

*Missing values: respiratory rate (n=1), oxygen saturation (n=5), wheeze (n=7), temperature (n=1), weight (n=5), height (n=14), prostration (n=30).
†Tachycardia defined as heart rate at admission >160 bpm in infants 0–11 months, >150 bpm in children 12–23 months, >140 bpm in children 2–4 years and >100 bpm in those aged ≥5 years.
‡Prostration defined as inability to sit if usually able or inability to feed.
§PCV doses if age 2–59 months; PCV7 only (no consolidation (n=441), consolidation no lung or pleural aspirate (LA/PA (n=156)), consolidation LA/PA (n=58)), PCV13 only (no consolidation (n=300), consolidation no LA/PA (n=88), consolidation LA/PA (n=6)), PCV7 and PCV13 (no consolidation (n=195), consolidation no LA/PA (n=65), consolidation LA/PA (n=17)).
BMI, body mass index; PCV, pneumococcal conjugate vaccine; WfH, weight for height.

infections were detected in 18 (11%) specimens with bacterial–viral coinfections in 30 (19%).

The most frequent pathogens by multiplex PCR in lung specimens were *S. pneumoniae* (n=68, 43%), *S. aureus* (n=26, 16%), Hib (n=11, 7%), bocavirus (n=11, 7%), influenza viruses (n=11, 7%), *Pneumocystis jirovecii* (n=10, 6%), *Moraxella catarrhalis* (n=8, 5%), *Salmonella* spp (n=8, 5%) and parainfluenza virus (PIV) 1 (n=8, 5%) (table 2). Respiratory syncytial virus (RSV) was detected in only three specimens. *S. pneumoniae* was more prevalent in patients aged ≥2 years (42/83, 51%) compared with children aged 0–23 months (26/77, 34%), OR 2.01 (95% CI 1.01 to 4.01). In contrast, *S. aureus* was more common in children aged <5 years (22/120, 18%) compared with older children and adults(4/40, 10%), OR 2.02 (95% CI 0.62 to 8.58). Hib was restricted to children aged <5 years. *P. jirovecii* was more common in children aged 0–23 months (8/77, 10%) compared with patients aged ≥5 years (2/83, 2%), OR 4.75(95% CI 0.90 to 47.0).

Coinfection by pairs of pathogens is shown in table 3. *M. catarrhalis* was detected in eight patients and in every case there was coinfection with *S. pneumoniae* (8/68 with *S. pneumoniae* vs 0/92 without *S. pneumoniae*, p=0.0007). *B. pertussis* was detected in seven patients and in six there was coinfection with *S. pneumoniae* (6/68 with *S. pneumoniae* vs 1/92 without *S. pneumoniae*, p=0.018). These comparisons are subject to multiple testing of 54 pairs of pathogens.

Using lung aspirate PCR results, the proportion of children aged 2–59 months hospitalised with clinical pneumonia in whom *S. pneumoniae* was detected was lower among those who had received≥2 doses of PCV compared with zero doses (table 4); age stratified OR 0.42(95% CI 0.16 to 1.05). Using a combination of culture and lung specimen PCR results, the proportion in whom *S. pneumoniae* was detected was less among those who had received ≥2 doses of PCV compared with zero doses (online supplemental table 2); age-stratified OR 0.54 (95% CI 0.33 to 0.90). Using culture and serotyping results, the

**Table 2** Organisms identified by multiplex PCR assay in patients with lung (n=156) and pleural (n=4) aspirate specimens

| Specific pathogens isolated | 0–23 months (N=77) n (%) | 2–4 years (N=43) n (%) | ≥5 years (N=40) n (%) | All ages (N=160) n (%) |
|---|---|---|---|---|
| *Streptococcus pneumoniae* | 26 (34) | 22 (51) | 20 (50) | 68 (42.5) |
| *Staphylococcus aureus* | 15 (19) | 7 (16) | 4 (10) | 26 (16.3) |
| *Haemophilus influenzae* type b | 6 (8) | 5 (12) | 0 (0) | 11 (6.9) |
| *Pneumocystis jirovecii* | 8 (10) | 1 (2) | 1 (3) | 10 (6.3) |
| *Moraxella catarrhalis* | 3 (4) | 4 (9) | 1 (3) | 8 (5.0) |
| *Salmonella* species | 5 (6) | 1 (2) | 2 (5) | 8 (5.0) |
| *Bordetella pertussis* | 3 (4) | 3 (7) | 1 (3) | 7 (4.4) |
| *Haemophilus influenzae* non-type b | 2 (3) | 3 (7) | 1 (3) | 6 (3.8) |
| *Chlamydia pneumoniae* | 0 (0) | 2 (5) | 1 (3) | 3 (1.9) |
| *Mycoplasma pneumoniae* | 1 (1) | 0 (0) | 1 (3) | 2 (1.3) |
| Bocavirus | 7 (9) | 1 (2) | 3 (8) | 11 (6.9) |
| Parainfluenza 1 | 3 (4) | 3 (7) | 2 (5) | 8 (5.0) |
| Influenza C | 2 (3) | 3 (7) | 2 (5) | 7 (4.4) |
| Cytomegalovirus | 4 (5) | 2 (5) | 0 (0) | 6 (3.8) |
| Coronavirus HKU1 | 2 (3) | 0 (0) | 2 (5) | 4 (2.5) |
| Coronavirus 43 | 0 (0) | 4 (9) | 0 (0) | 4 (2.5) |
| Respiratory syncytial virus | 2 (3) | 1 (2) | 0 (0) | 3 (1.9) |
| Influenza A | 2 (3) | 0 (0) | 0 (0) | 2 (1.3) |
| Influenza B | 1 (1) | 0 (0) | 1 (3) | 2 (1.3) |
| Rhinovirus | 1 (1) | 0 (0) | 1 (3) | 2 (1.3) |
| Adenovirus | 1 (1) | 1 (2) | 0 (0) | 2 (1.3) |
| Human metapneumovirus | 2 (3) | 0 (0) | 0 (0) | 2 (1.3) |
| Pathogen(s) isolated | | | | |
| Any pathogen | 52 (68) | 35 (81) | 29 (73) | 116 (72.5) |
| No pathogen | 25 (32) | 8 (19) | 11 (27) | 44 (27.5) |
| One pathogen | 25 (32) | 16 (37) | 20 (50) | 61 (38.1) |
| Two pathogens | 14 (18) | 14 (33) | 5 (10) | 33 (20.6) |
| Three pathogens | 9 (12) | 1 (2) | 3 (8) | 13 (8.1) |
| Four or more pathogens | 4 (5) | 4 (9) | 1 (3) | 9 (5.6) |
| Bacterial pathogen(s) | 43 (56) | 30 (70) | 24 (60) | 97 (60.6) |
| Bacterial pathogen(s) only | 30 (39) | 20 (47) | 17 (43) | 67 (41.9) |
| Viral pathogen(s) | 23 (30) | 15 (35) | 11 (28) | 49 (30.6) |
| Viral pathogen(s) only | 9 (12) | 5 (12) | 4 (10) | 18 (11.3) |
| Coinfections isolated | | | | |
| Bacterial–bacterial codetection | 11 (14) | 9 (21) | 6 (15) | 26 (16.3) |
| Bacterial–viral codetection | 13 (17) | 10 (23) | 7 (18) | 30 (18.8) |
| Viral-viral codetection | 6 (6) | 2 (5) | 0 (0) | 7 (4.4) |

*H. influenzae* non-type b if *H. influenzae* target positive and Hib target negative; Hib if both targets positive.

proportion of children in whom vaccine-type pneumococci were isolated was significantly less among those who had received ≥2 doses of PCV compared with zero doses table 4); age-stratified OR 0.17 (95% CI 0.06 to 0.51).

The greatest pathogen load in lung specimens was associated with *S. pneumoniae* (median 5.34 (IQR 3.73–6.24) $\log_{10}$ copies/ml), *H. influenzae* non-type b (median 6.07 (IQR 5.32–6.86) $\log_{10}$ copies/mL) and PIV 1 (median 6.46 (IQR 4.74–10.93) $\log_{10}$ copies/mL) positive specimens (online supplemental table 1). Low pathogen load was associated with *S. aureus* (median 2.15 (IQR 1.68–4.14) $\log_{10}$ copies/mL), bocavirus (median 2.77 (IQR

**Table 3** Frequency of detection of pairs of pathogens identified by multiplex PCR assay in 156 lung and 4 pleural aspirate specimens

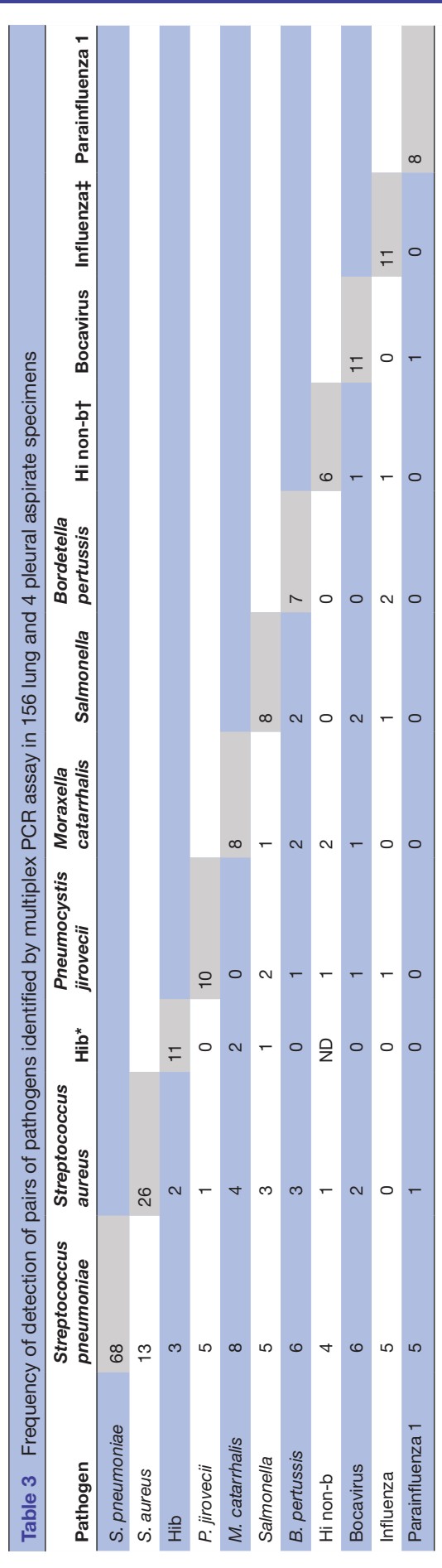

| Pathogen | Streptococcus pneumoniae | Streptococcus aureus | Hib* | Pneumocystis jirovecii | Moraxella catarrhalis | Salmonella | Bordetella pertussis | Hi non-b† | Bocavirus | Influenza‡ | Parainfluenza 1 |
|---|---|---|---|---|---|---|---|---|---|---|---|
| S. pneumoniae | 68 | | | | | | | | | | |
| S. aureus | 13 | 26 | | | | | | | | | |
| Hib | 3 | 2 | 11 | | | | | | | | |
| P. jirovecii | 5 | 1 | 0 | 10 | | | | | | | |
| M. catarrhalis | 8 | 4 | 2 | 0 | 8 | | | | | | |
| Salmonella | 5 | 3 | 1 | 2 | 1 | 8 | | | | | |
| B. pertussis | 6 | 3 | 0 | 1 | 2 | 2 | 7 | | | | |
| Hi non-b | 4 | 1 | ND | 1 | 2 | 0 | 0 | 6 | | | |
| Bocavirus | 6 | 2 | 0 | 1 | 1 | 1 | 0 | 1 | 11 | | |
| Influenza | 5 | 0 | 0 | 1 | 0 | 1 | 2 | 1 | 0 | 11 | |
| Parainfluenza 1 | 5 | 1 | 0 | 0 | 0 | 0 | 0 | 0 | 1 | 0 | 8 |

*Hib—H. influenzae type b.
†Hi non-b—non-type b H. influenzae.
‡Influenza—influenza A, B, C.
ND, not determined.

2.19–3.40) $\log_{10}$ copies/mL)), and cytomegalovirus (2.57 (IQR 2.38–3.71) $\log_{10}$ copies/mL) positive specimens.

## DISCUSSION

We have investigated the aetiology of lobar pneumonia in rural West Africa by applying multiplex molecular methods to a large number of lung specimens. Pathogens were detected in 73% of specimens with bacteria predominant. *S. pneumoniae* (43%) was the dominant pathogen followed by *S. aureus* (16%). Coinfection was common (34%) with bacterial–bacterial coinfection similar in prevalence to bacterial–viral coinfection. We observed correlated coinfection between *M. catarrhalis* and *S. pneumoniae*. The estimated effectiveness of ≥2 doses of PCV to prevent vaccine-type pneumococcal pneumonia was 83% (95% CI 49% to 94%). We have shown previously the association of the pneumococcus with severe lobar pneumonia in the study area.[13 20] Despite a well-established vaccination programme, Hib was aetiologic in 9% of lobar pneumonia in young children. These cases may relate to disease before the age of immunisation, delayed vaccine administration, waning immunity or unvaccinated migrants, but also continued transmission despite over 91% coverage of the three-dose schedule.[21] Although ongoing cases of culture-positive invasive Hib disease are documented in the Gambia,[21] it is only our attention to non-bacteraemic pneumonia that revealed this type of residual Hib disease.

The finding of *S. aureus* aetiology in 18% of lobar pneumonia cases in young children is of concern given that empiric therapy for severe pneumonia in our setting is penicillin/ampicillin and gentamicin,[22] which has suboptimal activity against staphylococcus. Ceftriaxone is recommended for severely ill children with hypoxia, heart failure or who are unable to feed. Cloxacillin is recommended if no improvement in 48 hours or staphylococcal pneumonia is suspected.[22] Unfortunately, clinical features indicative of staphylococcal pneumonia are not reliable and radiology and microbiology are not generally available. A review by Scott *et al* of 33 studies from 1918 to 1997, based on LA culture, reported *S. aureus* in 15% of cases.[23] Studies analysing lung specimens using molecular methods report *S. aureus* in 3/53 specimens in the Gambia[9] and 7/37 specimens in PERCH.[2] The finding of *P. jirovecii* in 10% of lobar pneumonia in 0–23 month-olds was surprising as HIV prevalence is low in our setting. This relatively high prevalence may relate to undiagnosed HIV, HIV exposure, malnutrition or be related to chance with small numbers of cases (n=10). Additional data are needed before a recommendation for HIV testing in children with lobar pneumonia is considered in this setting. We found *M. catarrhalis*, *Salmonella* spp, *B. pertussis* and non-type b *H. influenzae* aetiologic in 4%–5% of cases of lobar pneumonia.

We did not expect to find bocavirus as the most prevalent virus associated with lobar pneumonia (11/160), although our data are consistent with parainfluenza and

**Table 4** Association of pneumococcal pneumonia with PCV vaccination status

| Pneumonia aetiology by PCR on lung/pleural aspirate | No of PCV doses (PCV7 or PCV13) | | Total N | Odds ratio (95% CI) |
|---|---|---|---|---|
| | ≥2 doses | 0 doses | | |
| Age 2–11 months | N=27 | N=11 | | |
| Pneumococcal PCR +ve | 4 | 4 | 8 | |
| Pneumococcal PCR –ve | 23 | 7 | 30 | 0.30 (0.04 to 2.16) |
| Proportion pneumococcal PCR +ve | 0.15 | 0.36 | 38 | |
| Age 12–23 months | N=26 | N=3 | | |
| Pneumococcal PCR +ve | 12 | 2 | 14 | |
| Pneumococcal PCR –ve | 14 | 1 | 15 | 0.43 (0.007 to 9.5) |
| Proportion Pneumococcal PCR +ve | 0.46 | 0.66 | 29 | |
| Age 2–4 years | N=18 | N=23 | | |
| Pneumococcal PCR +ve | 7 | 13 | 20 | |
| Pneumococcal PCR –ve | 11 | 10 | 21 | 0.49 (0.12 to 2.03) |
| Proportion Pneumococcal PCR +ve | 0.39 | 0.57 | 41 | |
| Combined age strata 2–59 months, M-H age-stratified OR=0.42 (95% CI 0.16 to 1.05)*, p=0.062† | | | | |
| Pneumonia aetiology by culture of blood or lung/pleural aspirate and pneumococcal serotyping | | | | |
| Age 2–11 months | N=540 | N=184 | | |
| Vaccine-type pneumococcal‡ | 1 | 1 | 2 | |
| Not vaccine-type pneumococcal | 539 | 183 | 722 | 0.34 (0.004 to 26.8) |
| Proportion vaccine-type pneumococcal | 0.002 | 0.005 | 700 | |
| Age 12–23 months | N=515 | N=81 | | |
| Vaccine-type pneumococcal‡ | 3 | 2 | 5 | |
| Not vaccine-type pneumococcal | 512 | 79 | 591 | 0.23 (0.03 to 2.82) |
| Proportion vaccine-type pneumococcal | 0.006 | 0.025 | 596 | |
| Age 2–4 years | N=230 | N=218 | | |
| Vaccine-type pneumococcal‡ | 2 | 13 | 15 | |
| Not vaccine-type pneumococcal | 228 | 205 | 427 | 0.14 (0.02 to 0.62) |
| Proportion vaccine-type pneumococcal | 0.009 | 0.059 | 441 | |
| Combined age strata 2–59 months, M-H age-stratified OR=0.17 (95% CI 0.06 to 0.51)*, p=0.0005† | | | | |

*Mantel-Haenzel age-stratified OR.
†Fisher's exact p value.
‡Vaccine-type defined as PCV7 serotypes for children who received PCV7, and PCV13 serotypes for children who received PCV13 or a combination of PCV7 and PCV13.
PCV7, 7-valent pneumococcal conjugate vaccine.

influenza viruses causing severe lower respiratory infections. The PERCH study found RSV to be the virus most associated with severe pneumonia, and bocavirus as the seventh most associated virus.[2] However, bocavirus is a documented cause of pneumonia in the Gambia[9] and South Africa.[4] The single-site nature of our study or variable seasonal transmission during the relatively short study period that included one wet season (typically the RSV season) and two dry seasons (typically low viral transmission) may explain the differences in the prevalence of bocavirus and RSV in our lung specimens. Delayed processing of specimens may relate to preferential detection of DNA (eg, bocavirus) compared with RNA (eg, RSV) viruses but this is unlikely given our close attention to specimen handling. Alternatively, the consistency of our data with the similar paucity of RSV detected in lung specimens in PERCH[2] and by Howie et al,[9] suggest

that differing mechanisms of disease may explain the low prevalence of RSV in lung specimens, with RSV causing primarily upper respiratory and bronchiolar infection without alveolar consolidation, and bocavirus causing parenchymal disease.

Our finding that bacteria dominate the aetiology of lobar pneumonia aligns with both historical studies using lung aspirates[5 7 8 24] and recent studies using lung aspirates and molecular detection methods.[2 9] A Gambian study from 2007 to 2009 investigated 53 lung and pleural aspirates and found *S. pneumoniae* in 48, *H. influenzae* in 12, *S. aureus* and *Acinetobacter* spp in three each and only one virus only infection. RSV, adenovirus and bocavirus were detected in coinfection in two cases each.[9] PERCH data from 2012 to 2013, in which PCR detected pathogens in 43% of 37 lung and 15 pleural aspirates, detected pneumococcus in 13 specimens, *S. aureus* in 7, Hib in

four, *M. catarrhalis* in 4, viruses in 3 and no RSV.[2] The predominance of bacteria in our data and the lung specimens of the PERCH study[2] and Howie *et al*,[9] which differs from the viral preponderance in the hospitalised pneumonia cases in PERCH, appears to be related to a difference in clinical phenotype, with PERCH cases having WHO-defined endpoint consolidation and/or infiltrates on radiograph. Interestingly, the definition of cases in the GABRIEL study specified the presence of WHO-defined endpoint consolidation (excluding cases with infiltrates only) and found a population attributable fraction of 42% for *S. pneumoniae*.[3] This value is consistent with reductions in radiological pneumonia hospitalisations following the introduction of PCV in many countries.[25–28]

Our observation of coinfection with two (21%), three (8%) and four or more pathogens (6%) underscores the polymicrobial nature of lobar pneumonia. Bacterial–bacterial and bacterial–viral coinfections were of similar prevalence. In the setting of coinfection, the estimation of aetiological proportions due to individual pathogens remains a challenge with all aetiological pathogens necessarily contributing to more than 100% of cases. The importance of coinfections, temporal pathogenesis and the interplay of viral upper and bacterial lower respiratory infections, raises the potential for vaccine interventions to impact pathogenesis involving non-target pathogens. The synergistic role of *S. pneumoniae* has already been demonstrated in a vaccine probe study showing the administration of PCV prevented hospitalisation with viral-associated lower respiratory disease.[29]

The correlation we observed between *M. catarrhalis* and *S. pneumoniae* is intriguing. This may be explained by true synergism or by correlation alone given these organisms commonly cocolonise the upper respiratory tract. Aspiration of upper respiratory flora in the pathogenesis of lobar pneumonia would result in codetection of such bacteria in lung tissue, if bacteria were able to avoid neutrophil killing and other clearance mechanisms.

We estimated the effectiveness of PCV against non-bacteraemic pneumococcal pneumonia, which has not been possible in most trials. Among adults in the Netherlands the efficacy of one dose of PCV13 was 45% to prevent non-invasive vaccine-type pneumococcal pneumonia and 75% to prevent vaccine-type invasive disease. Our estimates of PCV effectiveness against vaccine-type (OR 0.17; 95% CI 0.06 to 0.51) and all pneumococcal pneumonia (OR 0.42; 95% CI 0.16 to 1.05) are similar to the Gambian PCV9 trial estimates of efficacy against lung aspirate positive vaccine-type (73%) and all pneumococcal pneumonia (68%).[30]

The main strength of our study is the inclusion of a significant number of lung aspirate specimens combined with a sensitive and specific multiplex PCR assay. Our study was limited by aetiological testing of only 160/741 patients with lobar pneumonia, the limited range of potential pathogens detected and the limited sample size. The multiplex assay excluded measles and *M. tuberculosis*. The PERCH study found no cases of *M. tuberculosis* in lung

or pleural aspirates but it was isolated in The Gambia in 7/255 induced sputum specimens.[2] The already cited Gambian study of 53 lung specimens found no cases of *M. tuberculosis*.[9] Our analyses excluded *Legionella* and *Klebsiella* spp due to poor assay specificity. We were unable to detect the aetiological pathogen(s) in 28% of patients with a lung aspirate. Our cross-sectional design was not able to investigate the temporal aspects of pneumonia pathogenesis. The limited duration of our study may also introduce potential bias due to variation in the seasonal transmission of individual pathogens.

Understanding the contribution of less prevalent pathogens in lobar pneumonia, the age distribution of pathogen aetiology and questions concerning coinfection and synergism will require larger sample sizes. More sensitive and specific multiplex assays may identify additional pathogens. Studies of pneumonia aetiology, and childhood pneumonia in general, should carefully consider the use of specific case definitions, for example, separating bronchiolitis and lobar pneumonia phenotypes, to avoid heterogeneity in outcome measurements.[31] Longitudinal studies of pneumonia pathogenesis, or vaccine probe studies(such as with an RSV vaccine), may help determine the relationships between viruses and bacteria. Studies of pathogen gene expression in the lung[32] may reveal new therapeutic approaches.

Our study provides important information concerning the aetiology of lobar pneumonia in a setting with significant child mortality during the period of introduction of PCV. Our findings may not be generalisable to settings with different levels of vaccine coverage and nasopharyngeal bacterial carriage. Further studies using lung aspirates will address a number of remaining important questions.

**Author affiliations**
[1]Disease Control and Elimination, Medical Research Council Unit The Gambia at London School of Hygiene & Tropical Medicine, Fajara, Gambia
[2]Infection and Immunity, Murdoch Children's Research Institute, Melbourne, Victoria, Australia
[3]Faculty of Infectious & Tropical Diseases, London School of Hygiene & Tropical Medicine, London, UK
[4]Cumming School of Medicine, University of Calgary, Calgary, Alberta, Canada
[5]Centre for International Health, University of Otago, Dunedin, New Zealand

**Acknowledgements** We are grateful to the staff at Basse District Hospital and Bansang Hospital and staff of the Pneumococcal Surveillance Project who provided clinical evaluation and care for the patients. The Regional Health Teams in Upper and Central River Region provided logistic support.

**Contributors** GAM conceived and designed the study, conducted the analysis, wrote the first draft of the manuscript and acts as guarantor for the publication. JM and EM conducted multiplex qPCR analyses and reviewed the manuscript. MN, JP, AF, BA and IH enrolled the patients, collected the specimens and reviewed the manuscript. AM conducted conventional microbiological analyses and reviewed the manuscript. BG and PH advised on analysis and reviewed the manuscript. All authors approved the final version of the manuscript for submission.

**Funding** This work was supported by the Bill & Melinda Gates Foundation (grant number OPP1020327) and the Medical Research Council Unit The Gambia at London School of Hygiene and Tropical Medicine.

**Competing interests** None declared.

**Patient consent for publication** Not applicable.

**Ethics approval** Ethical approval was granted for the study by the Gambia Government/Medical Research Council (UK) Joint Ethics Committee (numbers 1087 and 1247). Written informed consent was obtained from patients or the parent/guardian for all study procedures. A separate written informed consent was obtained prior to each lung aspiration procedure.

**Provenance and peer review** Not commissioned; externally peer reviewed.

**Data availability statement** Data requests may be submitted via the MRCG Data Management and Archives department to the MRCG Scientific Coordinating Committee and Gambia Government/MRCG Joint Ethics Committee. Deidentified patient data may be available.

**ORCID iD**
Grant Austin Mackenzie http://orcid.org/0000-0002-4994-2627

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
