## [Reviewer comments · BMJ Open]

ARTICLE DETAILS

TITLE (PROVISIONAL)	Aetiology of lobar pneumonia determined by multiplex molecular analyses of lung and pleural aspirate specimens in The Gambia: findings from population-based pneumonia surveillance
AUTHORS	Mackenzie, Grant; McLellan, Jessica; Machuka, Eunice; Ndiaye, Malick; Pathirana, Jayani; Fombah, Augustin; Abatan, Baderinwa; Hossain, Ilias; Manjang, Ahmed; Greenwood, B.; Hill, Philip

VERSION 1 – REVIEW

REVIEWER	Rhedin, Samuel Karolinska Institute, Department of Medical Epidemiology and Biostatistics
REVIEW RETURNED	06-Oct-2021

GENERAL COMMENTS	The study by Mackenzie et al presents microbiological findings of lung- and pleural aspirates collected from children with lobar pneumonia in the Gambia. Establishing etiology in pediatric pneumonia is challenging due to bacterial colonization of the upper respiratory tract and limited sensitivity of blood cultures. For that reason, samples obtained directly from the source of infection, as presented in the study, are extremely precious and provides valuable information to the field for an improved understanding of pneumonia etiology in children. The study is well written and my comments are mainly about the discussion and interpretation of the results. I also have some methodological concerns regarding the sample handling given the low number of detected viruses that needs to be discussed. Abstract - I would recommend clarifying your study aim and adding your finding of a low number of detected viruses to the results section. Strengths and limitations of this study - I do not see any limitations listed here. Please add the most important limitations of the study Aim P4, R47-51: "We studied these questions in rural Gambia during the introduction of pneumococcal conjugate vaccination (PCV), applying conventional and molecular methods to lung specimens." - Please clarify the aim that is currently not very clearly stated and does not mention the analysis of vaccine-effectiveness that appears to be a secondary aim of your study. Introduction
--

- Consider adding a clarifying sentence about the challenges when trying to establish etiology in childhood pneumonia (colonization of the upper respiratory tract, limited sensitivity of blood cultures, inability to produce sputum of good quality) to underscore the value of your samples that in most places are considered “too invasive”.

Methods

P6, R28-30: Total nucleic acid was extracted from a 200µl aliquot of lung and pleural aspirates (easyMAG, bioMérieux, France) with an internal control. Extracts were subjected to quantitative multiplex PCR (Fast-track Diagnostics Resp-33 kit, Sliema, Malta) for a panel of 33 respiratory bacteria, fungi, and viruses (see Supplementary Material) with internal positive, and negative controls.19”

- How and for who long were the samples stored before analysis? RT over day? -80C freezer?

Results

- Do you have clear respiratory infection seasons in the Gambia? Did you see any correlation of identified agent with regard to season?

The finding of *S. aureus* aetiology in 18% of lobar pneumonia cases in young children is of concern given that empiric therapy for severe pneumonia in our setting is penicillin/ampicillin and gentamicin,22 which has sub-optimal activity against staphylococcus

- How does this relate to other studies? Have you considered contamination from the skin as a potential explanation. Did you observe indications of treatment failure supporting *s. aureus* etiology?

Table 4

It was not perfectly clear to me what you meant with “PCR pneumococcal”. Rephrase?

Major comments

P17, R13-17: “We did not expect to find bocavirus as the most prevalent virus associated with lobar pneumonia (11/160), although our data are consistent with parainfluenza and influenza viruses causing severe lower respiratory...”

This is unexpected and a bit worrying. Bocavirus is a very stable DNA virus, whereas most other respiratory viruses are more or less instable RNA-viruses that quite rapidly degrade in room temperature, in particular viruses that can only be detected by molecular based methods (eg. metapneumovirus), which could suggest suboptimal sample handling before PCR-analysis. Please discuss a potential methodological explanation here (see comment above regarding sample handling). Or could it be explained by the fact that cases were mostly included outside the respiratory season?

	To conclude, I think the study presents valuable data and merits publication if the limitations including potential methodological issues with sample handling/contamination are more clearly addressed/discussed.
--	--

REVIEWER	Iwashita, Yoshiaki Shimane University School of Medicine, Emergency and Critical Care Medicine
REVIEW RETURNED	07-Dec-2021

GENERAL COMMENTS	This paper described the aetiology of pediatric lobar pneumonia by using the lung and pleural aspirates in the Gambia. Since it is very rare to take transthoracic lung aspirates to identify the causative microorganism of pneumonia, this article is highly interesting. However, there are some concerns especially about the safety of the technique and the interpretation of PCR. Major points:  • Did the authors take sputum culture from the patients? If sputum culture were taken, discuss the comparison of sputum culture and lung/pleural aspirates. If not taken, Why did you not perform a less invasive test than lung aspirates? • The PCR test replicates the DNA information of the microorganism, and the presence of DNA does not necessarily mean a current infection. In other words, is it possible that we are looking at dead bacteria or past infections? • Identification multiple organisms do not necessarily mean co-infection because PCR may detect past infection. Minor points: Methods  • This surveillance is a sub-analysis of the previous study, however, you should describe the inclusion criteria of this surveillance briefly. Results:  • The authors described "there were no complications following the lung aspiration procedures". Since the safety of the lung aspirates is vital to evaluate this report, please describe the predesigned definition of complication in this survey.
--

REVIEWER	Eklundh, Annika Karolinska Institute, Department of Global Public Health
REVIEW RETURNED	13-Dec-2021

GENERAL COMMENTS	Mackenzie et al have conducted a population-based pneumonia surveillance study with the purpose to determine the causes of lobar pneumonia in rural Gambia using lung-/pleural aspirates. 160 patients (the majority children < 5 years) were included and underwent lung-/pleural aspiration. The study concludes that lobar pneumonia in rural Gambia was caused primarily by bacteria. The study is well-written, and I only have a few comments. Abstract Please clarify the aim of the study. Strengths and limitations Please specify which are the strengths and which are the limitations of the study? It is a little unclear which the limitations are in your list. Introduction
--

	Please consider to further develop your thoughts on the diagnostic challenges of pneumonia etiology. Several previous studies on pneumonia etiology in children in the post-PCV era have indicated that viral etiology is more common than bacterial etiology. (Zar et al, Lancet Respir Med 2016) Also, "The PERCH study may have underestimated..." – could this instead indicate that there has been a shift towards viral etiology? (PERCH Study Group, Lancet, 2019) Please clarify the aim of the study. Methods Setting Could the study design be further described (Cross-sectional study)? Which is the study period? In the results section you refer to the "21-month study period" – between which dates? Patients and procedures The inclusion criteria (for suspected pneumonia) are the WHO pneumonia criteria, this should perhaps be written out/referred to (since you are referring to the WHO-criteria for radiological pneumonia)? Which were the contraindications for lung-/pleural aspirate? Ethical considerations Consider writing something about the ethical aspect of taking lung-/pleura aspirates from children (since a majority of the cases included in the study were children). Results Which is the 21-month study period? Discussion The finding of S. aureus aetiology in 18% of the cases – have you considered/discussed the possibility of contamination from the skin?
--	--

VERSION 1 – AUTHOR RESPONSE

Reviewer 1:

The aim of the study has been clarified in the abstract under the Objective heading as, 'To determine the causes of lobar pneumonia in rural Gambia.'

Page 3 of 9

The number of viral detections is already stated under the Results heading in the abstract as, 'Bacteria (n=97) were more common than viruses (n=49). This is what I understand the Reviewer requested.

The Reviewer states that no limitations are listed in the Strengths and limitations section after the abstract. This section has been rewritten and the points that include limitations listed below:

o Strength - population-based pneumonia surveillance collecting gold standard specimens directly from the infected lung to determine the aetiology of lobar pneumonia.

o Strength - multiplex real-time quantitative PCR to detect up to 31 pathogens in lung specimens.

o Limitation – multiplex PCR excluded *Legionella*, *Klebsiella*, and *Mycobacterium*

tuberculosis.

- o Limitation - failure to detect a pathogen in 28% of patients with a lung specimen.

- o Strength - specific aetiology results and accurate vaccination records allowed calculation of the effectiveness of pneumococcal conjugate vaccine to prevent non-bacteraemic pneumococcal pneumonia.

- o Strength and limitation - results are generalisable to patients with lobar pneumonia but not all patients with clinical pneumonia.

- The aim of the study has been revised in the final sentence of the Introduction, 'We aimed to determine the aetiology of lobar pneumonia and the effectiveness of pneumococcal conjugate vaccine (PCV) to prevent pneumococcal pneumonia in rural Gambia.'

- The Reviewer has asked us to consider adding information in the Introduction about the challenges in establishing the aetiology of childhood pneumonia. The 1st sentence in the Introduction now reads, 'The aetiology of childhood pneumonia is difficult to determine for a number of reasons: the upper respiratory tract is often colonised by pneumonia pathogens, a problem exacerbated with the use of overly sensitive molecular methods and also related to false positive detection of antigens in urine inability to produce sputum of good quality, and the difficulty obtaining a specimen from the alveolar space.'

- The Reviewer has asked about specimen handling before analysis. Lung specimens were collected between April 8, 2011 and July 17, 2012. Specimens were transported to the laboratory immediately after collection, and within 30 minutes, and stored at -80°C. Nucleic acid extraction and PCR assays were conducted in two batches, the first in November and December 2011 and the second in November and December 2012. The real-time PCR procedures, including the reverse transcription of viral RNA, were identical to the PERCH project, which was conducted in The Gambia around the same time. In fact, the staff and laboratory involved in analysing our specimens were the same as those who analysed the PERCH project specimens. The time between specimen collection and analysis was less for data reported here compared to the PERCH specimens in The Gambia, so the sensitivity of viral detection in our specimens will have been very similar to the PERCH project.

In response to the Reviewer's question, I have added information to the 8th and 9th

th

sentences in the Patients and procedures section within the Methods.

'Following written, informed consent, lung aspiration was performed by a clinician using aseptic technique with a 21 gauge needle and 5ml syringe with 1 ml of sterile saline with an aliquot inoculated on culture media. Specimens were immediately transported to the MRC Gambia, Basse laboratory, for preliminary analysis and stored at -80°C.

I have also added the following as the 1st sentence of the 2nd paragraph in the Laboratory procedures section in the Methods, 'Molecular analysis of lung specimens was conducted in two batches, in November/December 2011 and 2012, using the same methods, staff and laboratory as in the PERCH study in The Gambia.'

Finally, the following sentence has been added at the beginning of the Supplementary material in the Methods for multiplex PCR assay, 'The laboratory, staff, and methods used to analyse the specimens were the same to those used in the PERCH study which was also conducted in The Gambia at a similar time.'

□ The Reviewer has asked about the seasonality of respiratory disease in The Gambia. We generally see more bacterial pneumonia in the hot, dry season (February – May) and more viral circulation in the wet season (June - October). In the 8th paragraph of the Discussion we note the limited duration of the study (21 months from April 2011 to July 2012) as a limitation to describing variation in seasonal transmission of pathogens.

- The Reviewer notes the significant proportion of aetiology related to *S. aureus* and asks its relationship to other studies, potential contamination, and treatment failure. In a review of LA

studies, Scott et al. found 33 reports between 1918-1997 with *S. aureus* detected in 15% (n=329) of cases. In studies analysing lung specimens using molecular methods, Howie reported *S. aureus* in coastal Gambia in 3/53 specimens and PERCH found *S. aureus* in 7/37 specimens (7/14 in Gambia). McNally's study in South African children using blood cultures and lower respiratory specimens found *S. aureus* the 2nd most prevalent aetiology after *S. pneumoniae*, and the 5th most prevalent in children who failed empiric therapy. We were not surprised by this finding as in our own setting, *S. aureus* was the second most common blood culture pathogen in population-based surveillance for pneumonia, sepsis, and meningitis. We conducted an analysis to investigate potential contamination of blood cultures but found no evidence as the time to positivity of blood cultures was not associated with death or prolonged hospital stay. *S. aureus* is a well-established cause of pneumonia and skin contamination is also unlikely as the LA procedure is very brief and specimens were collected using sterile procedure. We do not have detailed data on the clinical course of the patients that might suggest treatment failure associated with *S. aureus* aetiology.

□ I have rephrased the row headings in Table 4 as 'Pneumococcal PCR +ve' and 'Pneumococcal PCR -ve' to make the meaning more explicit.

□ The Reviewer notes the unexpected detection of bocavirus as the most prevalent virus and raises a potential methodological explanation questioning the handling of samples or an issue of seasonality. We were surprised to find bocavirus as the most prevalent virus although its role as an aetiological pathogen in pneumonia is now clear. Bocavirus has been detected in 10% of childhood pneumonia in China (Ning 2017 Hum Vac Imm), found in bronchoalveolar specimens from children with pneumonia in China (Wang PLOS ONE 2020) and in another study found in 10% of lower respiratory specimens in China (Ji Virol J 2021). In South Africa bocavirus was detected in the upper respiratory tract of 13% of clinical pneumonia cases and 8% of controls (Zar LID 2016), however the use of induced sputum increased the detection of bocavirus by 26%. Our findings are consistent with PERCH and Howie's earlier study that report low prevalence of viruses in lung specimens
Page 5 of 9

compared to bacteria. Among the 37 lung and 15 pleural specimens in PERCH viruses were detected in only three (bocavirus, HMPV, and adenovirus in one each) and none with RSV. Howie's earlier LA study found viruses in only 11/53 specimens with bocavirus, RSV, and adenovirus each in 3/53 lung aspirates. Given the rapid delivery of specimens to the laboratory and immediate storage at -80°C, there is no evidence to support the potential explanation of delayed handling of specimens with loss of RNA viruses with preferential survival of DNA viruses.

The 4

th and 5th sentences of the 3

rd paragraph in the Discussion now elaborate further on

potential explanations for bocavirus being the most prevalent virus in our lung specimens, 'The single-site nature of our study or variable seasonal transmission during the relatively short study period that included one wet season (typically the RSV season) and two dry seasons (typically low viral transmission) may explain the differences in the prevalence of bocavirus and RSV in our lung specimens. Delayed processing of specimens may relate to preferential detection of DNA (e.g. bocavirus) compared to RNA (e.g. RSV) viruses but this is unlikely given our close attention to specimen handling. Alternatively, the consistency of our data with the similar paucity of RSV detected in lung specimens in PERCH2 and by Howie and colleagues,⁹ suggest that differing mechanisms of disease may explain the low prevalence of RSV in lung specimens, with RSV causing primarily upper respiratory and bronchiolar infection without alveolar consolidation, and bocavirus causing parenchymal disease.

Reviewer 2

□ The Reviewer asks if sputum specimens were not taken, and if not, why a less invasive test

than lung aspiration was not performed? We did not take sputum from the patients as most were children and unable to produce sputum. In the absence of a sputum specimen we did collect a less invasive specimen; we took blood cultures. Lung aspiration is a well-established and safe procedure in The Gambia (Ideh et. al. Int J Tuberc Lung Dis 2011).

□ The Reviewer questions the diagnostic nature of PCR detection of nucleic acids as not necessarily meaning current infection. Likewise, that PCR detection of nucleic acids may relate to past infection and so 'co-detection' may not necessarily mean 'co-infection'. This is an important and classical question in infectious disease diagnostics. The role of PCR in infectious disease diagnostics is now well-established. The detection of microbial nucleic acid belonging to a recognised pneumonia pathogen in a clinical sample collected from the site of alveolar lung consolidation in a patient with current symptoms, signs and radiological evidence of pneumonia is generally considered aetiologically diagnostic. Studies have not established whether pathogenic microbes may colonise the periphery of the healthy human lung in an asymptomatic manner and it seems likely that the peripheral lung alveolar tissue is sterile in the healthy individual. The lung aspirate specimen is collected from the peripheral lung beyond the bronchiolar airways from the site of active pneumonic disease of substantial size. The Reviewer's assertion that we may be detecting only dead bacteria or past infection does not align with the current clinical state of active pneumonic disease in the patient; the patient's status is not consistent with a past infection. Bacteria involved in alveolar consolidation disease are in a dynamic state of replication versus death as the immune response acts to clear the current infection. So, even if a proportion of the DNA detected derive from dead bacteria, the relatively early stage of the alveolar disease in which lung aspirates are usually taken means that active bacterial replication is ongoing at Page 6 of 9

that point in time, particularly as such alveolar consolidation typically requires 2-4 weeks to resolve. Thus, given the clinical and radiological status of our patients it is reasonable to accept that the detection of nucleic acid of known pneumonia pathogens is aetiologically diagnostic in the majority of cases and so our findings are valid.

□ The Reviewer requests additional information on the inclusion criteria used in patient enrolment. The 2nd, 3rd, and 4th sentences in the Patients and procedures section of the Methods now reads,

'All patients presenting to the nine health facilities in the study area were screened 7 days per week, 24 hours per day, using standardized methods to detect possible cases of pneumonia, septicaemia, meningitis, referral and clinical investigation.^{12;13} We used cross-sectional enrolment including all patients aged ≥ 2 months with suspected pneumonia.

Suspected pneumonia was defined as a history of cough or difficulty breathing with the presence of any one of the following: respiratory rate ≥ 40 or ≥ 50 per minute for children aged greater than or less than 12 months respectively, lower-chest-wall-in-drawing, nasal flaring, grunting, oxygen saturation $< 92\%$, dullness to percussion, bronchial breathing or crackles on auscultation.'

□ The Reviewer requests information about safety monitoring following lung aspiration in the study. Standard Operating Procedures for clinical monitoring following lung aspiration were to record oxygen saturation, respiratory rate, pulse rate, respiratory distress, and overall condition every 5 minutes for 15 minutes after the procedure looking for any sign of deterioration. Then review and record observations every 30 minutes for 2 hours looking for any change before returning to standard observations. There were no reports of postprocedure desaturation, worsening respiratory distress, or haemoptysis during the study.

Reviewer 3

□ As requested, the aim of the study has been clarified in the Objectives section of the abstract, 'To determine the causes of lobar pneumonia in rural Gambia.'

□ The Strengths and limitations section now makes clear which points are strengths and which are limitations.

- o Strength - population-based pneumonia surveillance collecting gold standard specimens directly from the infected lung to determine the aetiology of lobar pneumonia.
- o Strength - multiplex real-time quantitative PCR to detect up to 31 pathogens in lung specimens.
- o Limitation – multiplex PCR excluded Legionella, Klebsiella, and Mycobacterium tuberculosis.
- o Limitation - failure to detect a pathogen in 28% of patients with a lung specimen.
- o Strength - specific aetiology results and accurate vaccination records allowed calculation of the effectiveness of pneumococcal conjugate vaccine to prevent non-bacteraemic pneumococcal pneumonia.
- o Strength and limitation - results are generalisable to patients with lobar pneumonia but not all patients with clinical pneumonia.

□ The Reviewer suggests to further develop the discussion in the Introduction of the diagnostic challenges of pneumonia aetiology and also suggests that the PERCH study may

Page 7 of 9

indicate a shift towards viral aetiology. In response, a new 1st sentence of the Introduction

has been added, 'The aetiology of childhood pneumonia is difficult to determine for a number of reasons: the upper respiratory tract is often colonised by pneumonia pathogens, a problem exacerbated with the use of overly sensitive molecular methods, the inability to produce sputum of good quality, and the difficulty obtaining a specimen from the alveolar space.'

Review of the PERCH study and related studies do not indicate a shift towards viral aetiology but suggest that the findings were influenced by the case definition used and methodological issues. PERCH concluded that 5% of severe and 10% of very severe pneumonia were attributable to *S. pneumoniae* which is at odds with the approximately 40% reduction in radiologically confirmed hospitalised pneumonia seen in many countries following the introduction of PCV (Greenberg et al. Vaccine 2015; Pirez et al. PIDJ 2014; Hammitt et al. Lancet 2019; Mackenzie et al. Lancet ID 2021). PERCH cases had WHO defined endpoint consolidation or infiltrates (32% with wheezing, 53% no radiological consolidation, and 31% had RSV, suggesting many had bronchiolitis [with or without pneumonia]). International Classification of Diseases' criteria for pneumonia exclude clinical bronchiolitis and thus its inclusion in PERCH will have diluted the aetiology of pneumonia with viral causes of bronchiolitis. Enrolment of very severe (often bacterial) cases in PERCH may have been challenging in PERCH. Importantly, the predominant reliance of the aetiological modelling on the detection of upper respiratory pathogens may have biased towards differences in viral prevalence in cases and controls with lesser differences in the prevalence of common commensals such as *S. pneumoniae* and *H. influenzae*, and organisms detected in the upper respiratory tract may not reflect the pathogens present in the alveolar space.

To support the assertion in the Introduction that PERCH may have underestimated *S. pneumoniae* as a cause of severe pneumonia, and to clarify that the clinical phenotype determined by the case definition appears to be the main reason for differences between PERCH and our findings, the following has been added to the 4th paragraph in the

Discussion, 'The predominance of bacteria in our data and the lung specimens of the PERCH study² and Howie and colleagues,

9 which differs from the viral preponderance in the hospitalised pneumonia cases in PERCH, appears to be related to a difference in clinical phenotype, with PERCH cases having WHO-defined endpoint consolidation and/or infiltrates on radiograph. Interestingly, the definition of cases in the GABRIEL study specified the presence of WHO-defined endpoint consolidation (excluding cases with

infiltrates only) and found a population attributable fraction of 42% for *S. pneumoniae*.

3 This

value is consistent with reductions in radiological pneumonia hospitalisations following the introduction of PCV in many countries.

25-28

□ As suggested, the aim of the study has been further clarified in the final sentence of the Introduction, 'We aimed to determine the aetiology of lobar pneumonia and the effectiveness of pneumococcal conjugate vaccine (PCV) to prevent pneumococcal pneumonia in rural Gambia.'

□ According to the Reviewer's suggestion, I have given further description of the study design in the 2nd and 3rd sentences of the Patients and procedures section of the Methods.

Page 8 of 9

'All patients presenting to the nine health facilities in the study area were screened 7 days per week, 24 hours per day, using standardized methods to detect possible cases of pneumonia, septicaemia, meningitis, referral and clinical investigation.

12;13 We used cross-sectional enrolment including all patients aged ≥ 2 months with suspected pneumonia.'

□ The Reviewer requests specification of the study period in the Methods but the study period is already specified in the final sentence of the 2

nd paragraph in the Methods.

□ The criteria for suspected pneumonia are a modification of WHO criteria and this is now described in the 4th sentence of the Patients and procedures section in the Methods, 'Suspected pneumonia was defined using modified WHO criteria, as a history of

□ The Reviewer asks for the contraindications to lung aspiration, which are now given in the 8

th sentence in the Patients and procedures section of the Methods, '... there were no contraindications (post-measles pneumonia, pneumatoceles on radiograph, skin sepsis, or no written informed consent), ...'

□ The Reviewer asks us to consider writing about the ethical aspect of taking lung/pleural aspirates from children. In addition to the current wording in the Ethical considerations that describes that written informed consent was obtained from patients or the parent/guardian for all study procedure, the following sentence has been added, 'A separate written informed consent was obtained prior to each lung aspiration procedure.' The following sentence has also been added to the Laboratory procedures section in the Methods, 'Microbiological results were used to inform patient care.'

□ As requested by the Reviewer, the dates of the study period given in the Methods have now also been added to the 1st sentence in the Results.

□ The Reviewer requests consideration of potential contamination from the skin as a cause of *S. aureus* detection in 18% of cases. Reviewer 1 also expressed this concern and I replicate here what has been included earlier in response to Reviewer 1.

In a review of LA studies, Scott et al. found 33 reports between 1918-1997 with *S. aureus* detected in 15% (n=329) of cases. In studies analysing lung specimens using molecular methods, Howie reported *S. aureus* in coastal Gambia in 3/53 specimens and PERCH found *S. aureus* in 7/37 specimens (7/14 in Gambia). McNally's study (Lancet 2007) in South African children using blood cultures and lower respiratory specimens found *S. aureus* the 2nd most prevalent aetiology after *S. pneumoniae*, and the 5th most prevalent in children who failed empiric therapy. We were not surprised by this finding as in our own setting, *S. aureus* was the second most common blood culture pathogen in population-based surveillance for pneumonia, sepsis, and meningitis. We conducted an analysis to investigate potential contamination of blood cultures but found no evidence as the time to positivity of blood cultures was not associated with death or prolonged hospital stay. *S. aureus* is a well-established cause of pneumonia and skin contamination is also unlikely as

the LA procedure is very brief and specimens were collected using sterile procedure.
 Thank you for your consideration of these responses and the revised manuscript. I am ready to make any further modifications needed.

VERSION 2 – REVIEW

REVIEWER	Rhedin, Samuel Karolinska Institute, Department of Medical Epidemiology and Biostatistics
REVIEW RETURNED	01-Feb-2022

GENERAL COMMENTS	The authors have done a nice job revising the manuscript and I am happy with the revised version.
---

REVIEWER	Iwashita, Yoshiaki Shimane University School of Medicine, Emergency and Critical Care Medicine
REVIEW RETURNED	06-Feb-2022

GENERAL COMMENTS	The authors have responded appropriately to the reviewers' questions. This is an interesting study showing the aetiology of lobar pneumonia in children by detecting a lung aspirates. I have no additional comments to offer.
--